# Feedback Descent: Open-Ended Text Optimization via Pairwise Comparison

## Abstract

Current preference learning methods discard the rich explanations humans naturally provide when comparing examples, collapsing detailed feedback into binary signals. We introduce *Feedback Descent*, a framework that widens this information bottleneck by leveraging textual feedback to enable directed optimization in text space rather than weight space. We show that in-context learning can transform structured feedback into gradient-like directional information, enabling targeted edits of text artifacts such as prompts, code, and JSON. Unlike prior approaches that collapse judgments into single bits, our evaluators pair each comparison with textual feedback, which functions as high-bandwidth supervision. The iteration loop is done purely at inference time, without modifying any model weights, and is task-agnostic. We evaluate Feedback Descent on three diverse domains and find that it outperforms state-of-the-art prompt optimization (GEPA), reinforcement learning methods (GRPO, REINVENT), and even specialized graph-based molecular optimizers. In the DOCKSTRING molecule discovery benchmark, Feedback Descent identifies novel drug-like molecules surpassing the 99.9th percentile of a database with more than 200,000 compounds across six protein targets.

## 1 Introduction

A central goal of machine learning is building systems that can perform tasks that are difficult or impossible even for humans. Reinforcement learning is a powerful framework that accomplishes this goal, since it can optimize with respect to feedback on its own outputs, rather than relying on supervised examples of desired outputs. Indeed, recent language models have demonstrated impressive feats in domains like math and programming (OpenAI, 2024; DeepSeek-AI et al., 2025; Google DeepMind, 2025; Zhu et al., 2024) through a combination of reinforcement learning and text-based reasoning. Unfortunately, existing reinforcement learning frameworks are designed to learn from impoverished supervision signals, typically either scalar rewards or pairwise preference data, where each annotation conveys at most a single bit per pair. These bottlenecks discard information about *why* one behavior is better and *how* to improve—information available in environment feedback or easily elicited from humans during annotation.

Our goal is to widen this information bottleneck, i.e., significantly increase the information the system can extract per unit of experience (Silver & Sutton, 2025). Collecting more detailed feedback is straightforward, e.g., with brief rationales explaining preferences; the challenge is turning such feedback into measurable improvement. Because free-form feedback does not define a differentiable objective, it cannot directly drive weight updates via backpropagation. Our core idea is to use an *optimization loop in text space* rather than weight space: we leverage the in-context learning capabilities of language models to translate feedback into targeted edits of text artifacts (e.g., code, prompts, molecules, JSON configs, etc) that improve a final performance objective.

To that end, we introduce *Feedback Descent*, a framework for continual optimization in text space. At each iteration, we generate a new candidate artifact based on all previous feedback. We compare this candidate against the current best artifact, and the evaluator returns a preference along with textual feedback explaining the choice. If the candidate is preferred, it becomes the new best. Repeating this loop yields semantically local, feedback-aligned improvements that implement gradient-like steps in text space. See Fig. 1 for a conceptual illustration. We provide theoretical intuition for why Feedback Descent can be effective. Under appropriate assumptions about feedback quality

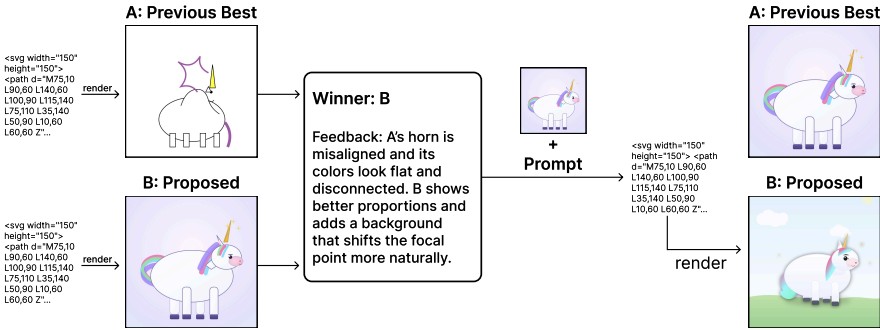

Figure 1: A conceptual illustration of feedback descent. At each iteration, we compare the previous best artifact with a new candidate. The evaluator provides both a pairwise preference and textual feedback. Preferences ensure the selection of better candidates, while feedback accumulates directional information that guides semantically meaningful edits.

and problem structure, we demonstrate that textual feedback can provide directional information, enabling efficient optimization.

Our contributions are threefold. First, we introduce *Feedback Descent*, an inference-time optimization framework that uses pairwise preferences with textual rationales to provide directional updates entirely in text space. Second, we demonstrate its generality across three domains: (i) SVG design, where iterative feedback produces judge-aligned visual improvements beyond direct prompting under both scratch and rubric-aware initializations; (ii) prompt optimization on IFBench, where Feedback Descent surpasses GEPA on Qwen3-8B and remains competitive with the strongest methods on GPT-4.1 Mini; and (iii) molecule discovery on DOCKSTRING, where Feedback Descent outperforms reinforcement learning approaches such as REINVENT and rivals specialized graph-based algorithms. Third, we show that the novel molecules discovered by Feedback Descent exceed not only the 99.9th percentile of a 260,000-compound database but, on several targets, surpass the best molecule present in the dataset.

## 2 FEEDBACK DESCENT: OPEN-ENDED TEXT OPTIMIZATION

We propose Feedback Descent, a framework for open-ended optimization of text-native artifacts whose quality is easier to *judge* than to *construct*. Feedback Descent converts comparative textual feedback into directed semantic edits and iterates in a self-improvement loop. As a running example, consider optimizing SVG code to render better images of a unicorn. Current vision-language models can reliably compare two renderings and explain the choice, even if writing high-quality SVG from scratch is difficult. Through Feedback Descent, we can convert these explanations into directed edits that aim to produce an artifact that is better than all previous ones.

### 2.1 PROBLEM SETUP

Let $\mathcal{S}$ be the space of token sequences, and let $x \in \mathcal{S}$ denote an artifact (e.g., SVG code). Given a current incumbent $x_t^\star \in \mathcal{S}$ and a candidate $x \in \mathcal{S}$, the evaluator returns

$$\mathsf{E}(x, x_t^\star) \to \big(p \in \{0, 1\}, \ r \in \mathcal{S}\big), \tag{1}$$

where $p = 1$ indicates $x \succ x_t^\star$ and $r$ is a textual feedback explaining *why* the winner is better and *how* to improve. We append $r_t$ to a history $\mathcal{R}_t = \{(x_1, r_1), \ldots, (x_t, r_t)\}$ and iterate, keeping track of the current best artifact $x_t^\star$.

### 2.2 FEEDBACK DESCENT

Feedback Descent operates as an iterative optimization loop that maintains a single best artifact $x_t^*$ and progressively improves it through feedback-guided mutations and comparative evaluation. Throughout, we use $\mathcal{M}$ to denote the language model used for generating improved candidates.

**Initialization and termination.** We initialize $x_0^*$ by prompting a language model with the task description alone (e.g., "Generate SVG code for a unicorn"), providing a reasonable starting point

without prior feedback. The algorithm runs for a fixed budget of $T$ iterations or until convergence (defined as no improvement for $k$ consecutive iterations).

**Proposing semantic mutations via prompting.** The mutation step leverages a language model's in-context learning capabilities. Given the current best artifact $x_t^*$ and accumulated feedback $r_1, r_2, \ldots, r_{t-1}$, we prompt the model to generate improved candidates:

$$x_t = \mathcal{M} \left( \text{"Improve } x_t^* \text{ using feedback: } \mathcal{R}_{t-1} \text{"} \right) \tag{2}$$

These prompts are intentionally minimal: the optimization signal comes from the accumulated feedback rather than heavy prompt engineering. They include basic task context, the current best artifact, and feedback from previous comparisons. Complete prompt templates for each domain are provided in Section C.3.

**Selection and update.** We compare the new candidate $x_t$ against the current best $x_t^*$ using the evaluator $\mathsf{E}(x_t, x_t^*)$, which returns both a binary preference $p_t$ and a textual feedback $r_t$. In our running SVG example, examples of feedback include "adjust the stroke width", "make sure the legs are connected to the body", and "add a shadow to the unicorn's mane". Regardless of the preference outcome, we always add the feedback to our history: $\mathcal{R}_{t+1} = \mathcal{R}_t \cup \{(x_t, r_t)\}$. If $p_t = 1$ (candidate is preferred), we update $x_{t+1}^* = x_t$; otherwise we keep $x_{t+1}^* = x_t^*$. We summarize the overall process in Algorithm 1.

---

**Algorithm 1** Feedback Descent

**Require:** Initial text $x_0$, Language model $\mathcal{M}, T$
1: Current best: $x^* \leftarrow x_0$, Rationale history: $\mathcal{R} \leftarrow \emptyset$
2: **for** $t = 1$ **to** $T$ **do**
3:      $x_t \leftarrow \mathcal{M}(x^*, \mathcal{R})$     ▷ Propose (2)
4:      $p_t, r_t \leftarrow \text{COMPARE}(x_t, x^*)$   ▷ Compare (1)
5:      $\mathcal{R} \leftarrow \mathcal{R} \cup \{(x_t, r_t)\}$
6:      **if** $p_t = 1$ **then**
7:          $x^* \leftarrow x_t, \mathcal{R} \leftarrow \emptyset$    ▷ Update + reset
8: **return** $x^*$

---

## 2.3 ANALOGY TO GRADIENT DESCENT

The key algorithmic insight is best understood by analogy to gradient descent. Just as gradients provide the direction of steepest ascent under local linearity, textual feedback can suggest plausible directions of improvement in semantic space. For our SVG example, if the feedback indicates "needs more defined horn shape," we expect that a small edit to the horn shape that preserves overall structure will likely be an improvement.

Of course, textual feedback is not a literal gradient. It is approximate and occasionally contradictory—optimization with such feedback does not have convergence guarantees in the same way that gradient descent does. Instead, feedback acts as a heuristic directional cue, offering higher-bandwidth supervision than a binary preference signal or a scalar reward, just as first-order optimization is fundamentally faster than zeroth-order optimization (Nemirovski & Yudin, 1983; Agarwal et al., 2012; Nesterov & Spokoiny, 2017). We hypothesize that an open-ended optimization loop based on such cues can succeed, supported by prior evidence that language models reliably translate textual instructions into concrete modifications. Examples include generating code changes (Chen et al., 2021; Austin et al., 2021; Nijkamp et al., 2022; Wang et al., 2023b; Roziere et al., 2023; Guo et al., 2024; Lozhkov et al., 2024; CodeGemma Team et al., 2024), following complex multi-step instructions (Ouyang et al., 2022; Wei et al., 2022a; Chung et al., 2022; Longpre et al., 2023; Zhang et al., 2024), targeted text modifications (Schick et al., 2022; Du et al., 2022; Madaan et al., 2023; Welleck et al., 2023; Kim et al., 2023), and decomposing high-level goals into executable action sequences (Schick et al., 2023; Parisi et al., 2022; Yao et al., 2023b; Qin et al., 2023; Wang et al., 2023a; Agarwal et al., 2025).

**Why directional information helps.** Zeroth-order methods that rely only on function evaluations or binary preferences suffer severe dimension-dependent slowdowns: convergence rates degrade exponentially as the search space grows (Nemirovski & Yudin, 1983; Nesterov & Spokoiny, 2017). In contrast, first-order methods exploit gradient information to achieve dimension-free convergence under standard assumptions. Textual feedback provides an approximation to such directional information. Even when individual rationales are imperfect, their aggregate message across failures continually refines the direction of improvement. We formalize this intuition in Section A, showing that under idealized assumptions, rationale-guided updates can achieve linear convergence rates independent of effective dimensionality, while zeroth-order baselines scale exponentially worse. These results provide motivation rather than rigorous guarantees for the discrete text domains we study

empirically. In Section 4, we show that Feedback Descent indeed produces consistent improvements across tasks, validating that such heuristic directional cues are sufficient to drive open-ended text optimization.

# 3 RELATED WORK

**Preference Learning.** Preference learning methods learn from pairwise comparisons (Christiano et al., 2017; Ouyang et al., 2022; Azar et al., 2023; Ethayarajh et al., 2024; Munos et al., 2024); recent advances include bypassing the need for a reward model (Rafailov et al., 2023), iterative optimization under KL constraints (Xiong et al., 2023), and adaptive scaling techniques (Wang et al., 2024). However, these methods fundamentally compress complex human reasoning into binary or scalar preferences, foregoing the rich explanatory content that humans can naturally provide alongside judgments (Wirth et al., 2017). Unlike prior work that relies solely on scalar feedback despite the complexity of human judgment, we leverage detailed textual rationales to widen this information bottleneck, allowing for more efficient adaptation.

**Evolutionary Algorithms and Gradient-Free Optimization.** Feedback Descent can be viewed as an evolutionary algorithm (Golberg, 1989; Holland, 1992), in which candidates are iteratively mutated and accepted based on fitness. While the black-box nature of modern LLMs has spurred interest in applying gradient-free approaches (Guo et al., 2023; Sun et al., 2022; Chen et al., 2024; Lange et al., 2024), these methods face fundamental challenges in high-dimensional spaces. More broadly, zeroth-order methods (Chen et al., 2019) face convergence rates that scale poorly with dimension, which is consistent with our experimental results comparing with reinforcement learning methods in Section 4. Feedback Descent explores whether textual rationales can provide useful directional information for optimization, similar to how Nie et al. (2024) shows that LLMs can be effective optimizers when provided with directional feedback from historical traces. Our contribution is in operationalizing an effective *directed mutation operator* via accumulated textual feedback.

**Optimizing Compound AI Systems.** Compound AI systems, i.e., modular architectures involving multiple LLM invocations and complex control flow, such as agents or scaffolding techniques (Yao et al., 2023b), present unique optimization challenges due to their modularity. Several approaches have emerged to tackle this complexity, including optimization for searching and bootstrapping few-shot in-context examples (Khattab et al., 2022; 2024; Opsahl-Ong et al., 2024), backpropagating textual feedback between components (Yuksekgonul et al., 2024), and reflective prompt evolution (Agrawal et al., 2025). However, these methods focus on optimizing individual components or connections within fixed architectures. In contrast, Feedback Descent provides a general-purpose text optimization framework that treats LLMs as optimizers for any text-representable artifact. While compound AI systems are one promising application domain, our approach generalizes beyond AI systems to optimize standalone text artifacts such as SVG code and molecular representations.

**Inference-Time Optimization for LLMs.** Inference-time optimization improves performance without weight updates by performing additional computation at generation. This paradigm includes self-critique and refinement cycles (constitution-guided critique (Bai et al., 2022); Self-Refine (Madaan et al., 2023)) test-time scaling via best-of-$N$, multi-step reasoning, and tree search (Cobbe et al., 2021; Zelikman et al., 2022; Yao et al., 2023a), and iterative prompt optimization (Zhou et al., 2022; Yang et al., 2023; Pryzant et al., 2023). Several works report that strategically allocating inference-time compute yields large gains (Snell et al., 2024; Brown et al., 2025; Geiping et al., 2025; Zhou et al., 2025). We build on the growing consensus that natural language is a particularly powerful medium for inference-time improvement. Natural language traces enable models to reason effectively in complex environments (Lampinen et al., 2022; Wei et al., 2022b), and language models can reliably map textual instructions to concrete modifications (Chen et al., 2021; Austin et al., 2021; Saunders et al., 2022; Scheurer et al., 2023). However, existing methods often rely on random sampling of self-generated critiques, which may be noisy or fail to capture external preferences. In contrast, we leverage external rationales as directional information, enabling guided search in the semantic space.

# 4 EXPERIMENTS

We evaluate Feedback Descent across three diverse domains—visual design, prompt optimization, and molecule discovery—to demonstrate its generality and effectiveness. Through our experiments,

| Model | Condition | Anatomy | Cyber | Geom | Min. | Retro | Story |
|-------|-----------|---------|-------|------|------|-------|-------|
| GPT-4o-mini | Scratch | $95.2 \pm 8.3$ | $97.6 \pm 4.1$ | $87.7 \pm 5.3$ | $100.0 \pm 0.0$ | $100.0 \pm 0.0$ | $100.0 \pm 0.0$ |
|  | Informed | $92.1 \pm 9.3$ | $91.2 \pm 4.8$ | $93.0 \pm 7.8$ | $92.8 \pm 7.7$ | $69.8 \pm 47.2$ | $95.6 \pm 3.8$ |
| GPT-5-mini | Scratch | $100.0 \pm 0.0$ | $100.0 \pm 0.0$ | $100.0 \pm 0.0$ | $100.0 \pm 0.0$ | $100.0 \pm 0.0$ | $100.0 \pm 0.0$ |
|  | Informed | $92.1 \pm 9.3$ | $89.9 \pm 7.0$ | $96.3 \pm 3.2$ | $95.9 \pm 3.6$ | $96.1 \pm 3.4$ | $100.0 \pm 0.0$ |

Table 1: Win rates after five iterations comparing Feedback Descent against direct prompting under two conditions: from *Scratch* and *Informed* of the judge rubric. We show means and standard deviations across 3 random seeds. **Iterative feedback consistently improves SVG designs over direct prompting.**

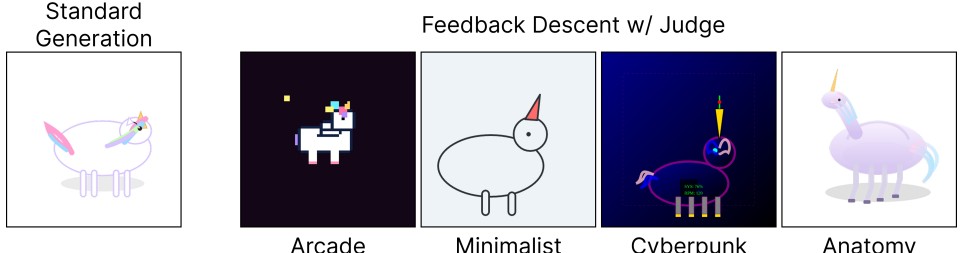

Figure 2: Example unicorn images generated by Feedback Descent under four different judge criteria: retro arcade, minimalist, cyberpunk, and anatomy. **Feedback Descent yields visually distinct unicorns aligned with the aesthetic criteria preferred by each judge.**

we aim to answer the following questions. First, we ask whether Feedback Descent exhibits generality by working robustly across qualitatively different domains. Second, we test sample efficiency, evaluating whether iterative, rationale-guided feedback enables higher-quality solutions with fewer model queries than existing optimizers. Third, we measure outcome quality, assessing whether Feedback Descent can produce artifacts (SVGs, prompts, and molecules) that not only satisfy rubrics and constraints but also surpass state-of-the-art methods on established benchmarks.

### 4.1 EXPERIMENTAL DOMAINS

We describe each evaluation domain and how we obtain pairwise comparisons augmented with textual rationales.

**SVG optimization.** Taking inspiration from Bubeck et al. (2023), we ask models to output SVG code for illustrations of unicorns. We use a set of six diverse judge prompts, each preferring a different aesthetic: accurate *anatomy*, *cyberpunk* futurism, *geometric* abstraction, *minimalist*, *retro arcade* pixel-art motifs, and *storybook* illustrations. We compare rendered SVGs using `GPT-5-mini`, which outputs both a binary preference and short textual feedback. To mitigate order bias, we perform two judgments with swapped image orders (A-B and B-A) and declare a winner only if both judgments are consistent. Otherwise, we try again, up to three times, and discard if no consistent winner emerges.

**Prompt optimization.** We follow the setup of GEPA (Agrawal et al., 2025) on IFBench (Pyatkin et al., 2025), a benchmark for evaluating precise constraint-following (e.g., "answer only with yes or no"). We design a two-stage system that first produces an answer and then rewrites it to satisfy constraints, and we jointly optimize the prompts for both stages using Feedback Descent. Optimization is driven by the 150 training examples: candidate prompts are updated based on performance on the training set and textual feedback describing which constraints were satisfied or violated. All candidate prompts are scored on the 300 validation examples, and the prompt with the highest validation accuracy rate is selected. We report performance on a test set of 294 held-out examples.

**Molecule discovery.** We evaluate on molecular docking tasks using DOCKSTRING (García-Ortegón et al., 2022) docking scores and drug-likeness (QED). DOCKSTRING provides a realistic drug discovery setting where molecules are evaluated based on their predicted binding affinity to medically relevant targets rather than simple physicochemical properties. We focus on challenging optimization tasks across six protein targets: ADRB1, PGR, PPARA, PPARG, CDK2, and F2. Following DOCKSTRING, we compute the combined score $s = -\text{Vina} - 10 \times (1 - \text{QED})$. We

| Method | Qwen3-8B | GPT-4.1 Mini |
|---|---|---|
| DSPy Default (Khattab et al., 2024) | 36.90 | 47.79 |
| MIPROv2 (Opsahl-Ong et al., 2024) | 36.22 | 49.15 |
| GRPO (Shao et al., 2024) | 35.88 | — |
| GEPA (Agrawal et al., 2025) | 38.61 | 52.72 |
| GEPA+Merge (Agrawal et al., 2025) | 28.23 | **55.95** |
| Ours | **44.22** $\pm$ 3.15 | 54.59 $\pm$ 2.46 |

Table 2: Comparison of prompt optimization methods on IFBench. We report scores for Qwen3-8B and GPT-4.1 Mini under matched rollout budgets. **Feedback Descent outperforms all baselines on Qwen3-8B, and is competitive with the state-of-the-art for GPT-4.1 Mini.**

represent molecules as SMILES strings (Weininger, 1988) and evaluate using DOCKSTRING's molecular docking pipeline to compute Vina scores (binding affinity). The feedback system provides rich structured information, including RDKit molecular descriptors (Landrum, 2006), similarity searches against known compounds from molecular databases (Liu et al., 2007; Gilson et al., 2016; Gaulton et al., 2012; Mendez et al., 2019), and detailed docking results. In the system prompt, we also provide the LLM information about the protein target obtained from the UniProt database (The UniProt Consortium, 2023). Together, this provides the LLM with detailed feedback on molecular properties that affect binding affinity, drug-likeness violations, and comparisons to known active compounds.

## 4.2 SVG OPTIMIZATION

We evaluate iterative feedback against direct prompting across two generators, `GPT-4o-mini` and `GPT-5-mini`. The direct prompting baseline receives the full evaluation rubric and is tasked with producing a single best design. Feedback Descent instead begins with an initial set of candidates, and through 5 rounds of structured feedback and improvement, refines designs using judge comparisons that reflect aesthetic criteria. We test two initialization regimes: **Scratch**, which starts from images simply instructed to generate images of unicorns, and **Informed**, which starts from the strongest direct generations conditioned on the rubric, determined by the LLM judge.

**Results.** Table 1 shows the win rates after 5 iterations. For both `GPT-4o-mini` and `GPT-5-mini`, Feedback Descent reliably improves outputs over the initial population. Furthermore, qualitative examples in Fig. 2 demonstrate that the procedure consistently produces unicorns whose visual style diverges across judges, aligning with aesthetic criteria such as geometry, minimalism, or retro arcade motifs.

> **Iterative feedback can elicit better outputs from the same model**
>
> Because of a generator–verifier gap, even prompting with the exact judge rubric is suboptimal for SVG generation. Feedback Descent elicits better images from the same generator by iteratively proposing improvements guided by feedback.

## 4.3 PROMPT OPTIMIZATION

We compare Feedback Descent against five baselines: the default prompt implemented in the DSPy program (Khattab et al., 2024, Default), a Bayesian optimization approach for selecting instructions and demonstrations (Opsahl-Ong et al., 2024, MIPROv2), online reinforcement learning (Shao et al., 2024, GRPO), and a reflective prompt evolution method (Agrawal et al., 2025, GEPA). All baselines are run under matched rollout budgets for fair comparison, and the reported baseline results are from Agrawal et al. (2025).

Each example produces pointwise feedback about which constraints were satisfied or violated. To construct the pairwise feedback for Feedback Descent, we stratify the examples into quadrants based on whether each prompt resulted in a correct response. We then ask the model to propose textual descriptions of inputs where these discrepancies arise. We then statistically validate each hypothesis, filtering for ones that correspond to consistent differences in performance between the prompts. This process distills the true global differences between the two prompts.

| | Method | ADRB1 | PGR | PPARA | PPARG | CDK2 | F2 |
|---|---|---|---|---|---|---|---|
| **DOCKSTRING** (N=260155) | Top 50% | 5.305 | 3.478 | 4.549 | 4.210 | 4.385 | 4.168 |
| | Top 90% | 8.785 | 7.878 | 7.987 | 7.658 | 7.733 | 7.477 |
| | Top 99% | 9.620 | 8.703 | 8.718 | 8.449 | 8.453 | 8.139 |
| | Top 99.9% | 10.209 | 9.260 | 9.230 | 9.012 | 8.979 | 8.722 |
| | Top 99.99% | 10.742 | 9.723 | 9.821 | 9.518 | 9.509 | 9.252 |
| | Best Molecule | 11.330 | 9.742 | 9.907 | 9.529 | 9.534 | 9.311 |
| | GP-BO[†] (Tripp et al., 2021) | 10.552 | 9.307 | 9.680 | 9.485 | 9.067 | 8.686 |
| | Graph MCTS[†] (Jensen, 2019) | 8.883 | 7.819 | 7.363 | 7.134 | 7.777 | 6.310 |
| | Graph GA[†] (Jensen, 2019) | 9.145 | 8.670 | 8.598 | 8.327 | 8.288 | 8.102 |
| | SMILES GA (Brown et al., 2019) | 9.334 | 8.335 | 9.052 | 8.560 | 8.268 | 7.984 |
| | REINVENT (Olivecrona et al., 2017) | 9.018 | 8.267 | 8.430 | 8.347 | 8.226 | 8.139 |
| | TextGrad (Yuksekgonul et al., 2024) | 8.531 | 8.057 | 7.953 | 7.256 | 8.174 | 7.357 |
| | Feedback Descent | **10.623** | **9.615** | **9.919** | **10.187** | **9.803** | **9.300** |

Table 3: Comparison of molecule optimization methods on six protein targets. Fragment-based algorithms (denoted by †) operate directly on molecular graphs, giving them structural priors unavailable to purely text-based methods. For each target, the top generative result is in **bold**, and any population in the DOCKSTRING that exceeds the best generative result is underlined. **Feedback Descent rivals or surpasses specialized molecular optimizers across all six targets.**

Table 2 shows that Feedback Descent achieves the highest score on Qwen3-8B (44.22 vs. 38.61 for GEPA) and remains competitive with GEPA and GEPA+Merge on GPT-4.1 Mini (54.59 vs. 55.95). These results indicate that structured, iterative feedback drives steady improvements in prompt optimization, even though other optimizers such as GEPA exploit problem structure.

> **Grounded Summaries Enable Reliable Prompt Optimization**
>
> By summarizing a large set of pointwise rationales into a global comparison between two prompts, Feedback Descent yields more reliable prompts.

## 4.4 MOLECULE OPTIMIZATION (DOCKSTRING)

We compare against baselines implemented in the `mol_opt` repository (Gao et al., 2022), Our comparisons include a genetic algorithm (Brown et al., 2019, SMILES GA), reinforcement learning (Olivecrona et al., 2017, REINVENT), fragment-based algorithms (Jensen, 2019, Graph MCTS/GA), and Bayesian optimization on molecular graphs (Tripp et al., 2021, GP-BO). Because fragment-based methods exploit graph-level structural priors, the most direct comparison is to the text-only baselines: SMILES-GA and REINVENT. Nonetheless, we report results against all methods to provide a complete picture of performance. **Results.** Table 3 summarizes optimization outcomes across six protein targets. For each target, we benchmark Feedback Descent against specialized molecular optimization algorithms as well as ligands from the DOCKSTRING dataset, which comprises both decoy and experimentally active ligands. Feedback Descent is competitive with all baselines and achieves the strongest scores on several targets (e.g., ADRB1, PGR, PPARG, CDK2, F2). On multiple proteins, it matches or exceeds the 99.9th and even 99.99th percentiles of the DOCK-

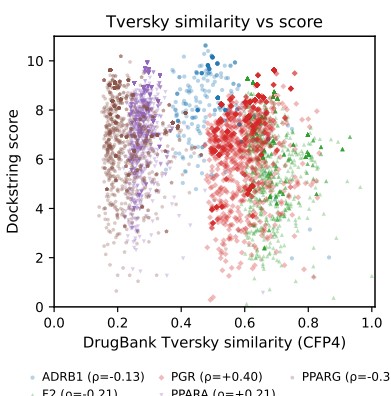

Figure 3: Scatter plots of Tversky similarity to approved drugs against docking scores, showing weak or negative correlations across targets. **High-scoring molecules discovered by Feedback Descent are far from any known drugs.**

STRING database, including surpassing the best molecule present in the dataset itself ($N = 260155$). These findings show that Feedback Descent, a purely text-based method, can rival or outperform specialized graph-based algorithms, despite lacking handcrafted structural priors. Fig. 4 shows optimization trajectories for PPARG. Feedback Descent achieves competitive trajectories relative to specialized methods, often reaching high-scoring regions of chemical space with comparable or fewer oracle calls. This pattern holds across targets, suggesting that the method generalizes rather than relying on idiosyncrasies of a single protein system.

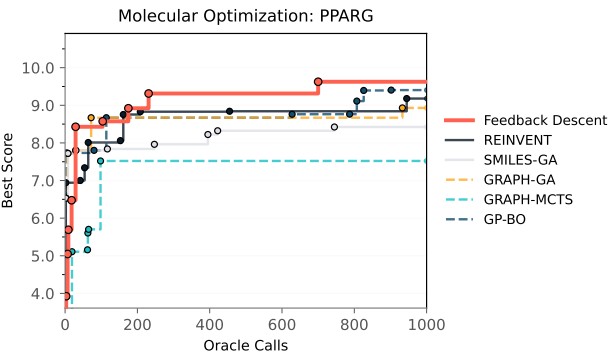

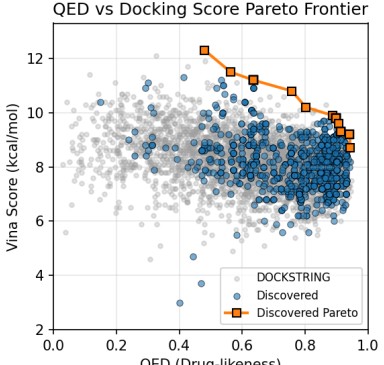

Figure 4: Optimization trajectories on PPARG showing docking scores over oracle calls for Feedback Descent and specialized baselines. **Feedback Descent quickly improves molecular docking scores within the first few hundred oracle calls.**

Figure 5: Pareto frontier of docking affinity vs. drug-likeness, comparing Feedback Descent molecules (blue) to the DOCKSTRING dataset (gray). **Feedback Descent finds novel molecules that meet or surpass known ones.**

**Analysis of discovered molecules.** Fig. 5 illustrates the Pareto frontier between docking affinity (Vina score) and drug-likeness (QED) for PPARG. Feedback Descent recovers molecules that sit on or above the DOCKSTRING frontier, indicating that improvements in affinity are not achieved at the expense of reduced drug-likeness. See Fig. 6 in the appendix for the full set of Pareto frontiers across all targets. These results show that feedback-guided search yields candidates that are not only potent but also balanced along multiple drug-relevant dimensions.

We also examine novelty by plotting Tversky similarity (CFP4 fingerprints) to approved DrugBank molecules against docking scores in Fig. 3. Across all targets, the correlations are weak or negative (Spearman $\rho$ between $-0.39$ and $0.40$), showing that high-scoring candidates discovered by Feedback Descent do not simply recycle functional groups from existing drugs but instead explore novel regions of chemical space. For CDK2, no comparison is shown: the target lacks any fully approved drugs in DrugBank with orthosteric binding as part of their mechanism of action, and thus does not satisfy our filtering criteria for inclusion.

> **Feedback Descent Can Discover Novel Targeted Molecules**
>
> Feedback Descent, operating in a purely textual form, consistently identifies novel molecules that surpass high-percentile baselines in DOCKSTRING. This demonstrates that iterative, feedback-guided optimization can enable models to genuinely explore unknown design spaces beyond their training distribution.

## 5 DISCUSSION

This paper presents Feedback Descent, an inference-time framework that improves text artifacts through structured pairwise feedback. We validate it on visual design, prompt optimization, and molecule discovery, showing that text can serve as an optimizable medium, not just static data. Unlike parameter tuning, this approach can leverage richer textual signals, allowing for continual improvement without requiring retraining.

**Limitations**. The method relies on strong evaluators, which may be scarce in some domains. Training models to produce reliable feedback remains a prerequisite for harder tasks. For creative domains, strictly "following the gradient" may be limiting; balancing refinement with exploration is an important next step.

ETHICS STATEMENT

This work adheres to the ICLR Code of Ethics. Our research focuses on improving preference learning methods through textual rationales, which have positive implications for AI alignment and human-AI collaboration. The methods developed could potentially be misused to optimize for harmful content; the same risk exists with any preference learning approach. Our contribution lies in making such optimization more efficient rather than enabling fundamentally new capabilities.

REPRODUCIBILITY STATEMENT

We are committed to ensuring the reproducibility of our results. Complete experimental details, including hyperparameters and evaluation protocols, are provided in the main text and appendix. All datasets used in our experiments are either publicly available or will be released upon publication. The proofs are presented with full detail in Section A with all assumptions clearly stated. Implementation details for Feedback Descent, including prompting strategies and in-context learning procedures, are documented in the appendix.

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

## A  FORMAL STATEMENTS AND PROOFS

**Proposition 1** (Linear convergence under PL with rationale-guided directions). *Let $r : Z \to \mathbb{R}$ be $L$-smooth and satisfy the $\mu$-PL condition (for maximization)*

$$\frac{1}{2}\|\nabla r(z)\|_2^2 \;\geq\; \mu\big(r(z^\star) - r(z)\big) \qquad \forall z \in Z.$$

*At iteration $t$, suppose a direction $v_t$ satisfies*

$$\mathbb{E}[\,v_t \mid z_t\,] \;=\; \alpha\,\nabla r(z_t), \qquad \mathbb{E}\big[\,\|v_t - \mathbb{E}[v_t \mid z_t]\|_2^2 \,\big|\, z_t\,\big] \;\leq\; \sigma^2\|\nabla r(z_t)\|_2^2,$$

*with constants $\alpha > 0$ and $\sigma \geq 0$, and define $\kappa_1 \triangleq \alpha^2 + \sigma^2$. Consider the update $z_{t+1} = z_t + \eta v_t$. If a constraint set $Z$ is present, assume $z_t + \eta v_t \in Z$ (i.e., the projection is inactive). With stepsize $\eta = \alpha/(L\kappa_1)$,*

$$\mathbb{E}\big[r(z^\star) - r(z_{t+1}) \,\big|\, z_t\big] \;\leq\; \Big(1 - \tfrac{\mu\,\alpha^2}{L\,\kappa_1}\Big)\,\big[r(z^\star) - r(z_t)\big].$$

*Unrolling yields*

$$\mathbb{E}[r(z^\star) - r(z_T)] \;\leq\; \Big(1 - \tfrac{\mu\,\alpha^2}{L\,\kappa_1}\Big)^T\big[r(z^\star) - r(z_0)\big],$$

*so $\epsilon$-accuracy is achieved in*

$$T \;=\; O\!\left(\frac{L(\alpha^2 + \sigma^2)}{\mu\,\alpha^2}\,\log\frac{1}{\epsilon}\right)$$

*iterations.*

*Proof.* $L$-smoothness gives the two-sided bound

$$r(z_t + \eta v_t) \;\geq\; r(z_t) + \eta\langle\nabla r(z_t), v_t\rangle - \tfrac{L}{2}\eta^2\|v_t\|_2^2.$$

Taking conditional expectation and using $\mathbb{E}[v_t|z_t] = \alpha\nabla r(z_t)$ and $\mathbb{E}\big[\|v_t\|_2^2 \,\big|\, z_t\big] \leq (\alpha^2 + \sigma^2)\,\|\nabla r(z_t)\|_2^2 = \kappa_1\|\nabla r(z_t)\|_2^2$,

$$\mathbb{E}[r(z_{t+1}) \mid z_t] \;\geq\; r(z_t) + \Big(\eta\alpha - \tfrac{L}{2}\eta^2\kappa_1\Big)\,\|\nabla r(z_t)\|_2^2.$$

By the PL inequality, $\|\nabla r(z_t)\|_2^2 \geq 2\mu\,[r(z^\star) - r(z_t)]$, so

$$\mathbb{E}[r(z^\star) - r(z_{t+1}) \mid z_t] \;\leq\; \Big(1 - 2\mu\eta\alpha + \mu L\eta^2\kappa_1\Big)\,[r(z^\star) - r(z_t)].$$

Choosing $\eta = \alpha/(L\kappa_1)$ makes the bracket equal to $1 - \mu\alpha^2/(L\kappa_1)$, yielding the claim. $\qquad\square$

### A.1  QUERY COMPLEXITY AND DIMENSION DEPENDENCE

**Dimension-Free Case.** When rationales provide full gradient information ($v_t \in \mathbb{R}^d$) at unit cost, the query complexity equals $T$ and is dimension-independent:

$$\text{Queries} = O\left(\frac{L(\alpha^2 + \sigma^2)}{\alpha^2\mu}\,\log\frac{1}{\epsilon}\right) \tag{3}$$

**Coordinate-Sparse Case.** Suppose each query reveals one coordinate of $\nabla r(z_t)$ chosen uniformly at random. Using the unbiased estimator $v_t = d\,(\partial_i r(z_t))\,e_i$ with $i \sim \text{Unif}([d])$ gives $\alpha = 1$, $\sigma^2 = d - 1$, and hence $\kappa_1 = d$ and stepsize $\eta = 1/(Ld)$. We have

$$T = O\!\Big(\frac{Ld}{\mu}\,\log\frac{1}{\epsilon}\Big), \qquad \text{Queries} = O\!\Big(\frac{Ld}{\mu}\,\log\frac{1}{\epsilon}\Big).$$

Equivalently, averaging $m$ independent coordinate queries per iteration yields $\sigma^2 = (d-1)/m$; taking $m = d$ recovers $T = O((L/\mu)\log(1/\epsilon))$ with $d$ queries per iteration, so total queries remain $\Theta\!\big(\frac{Ld}{\mu}\log\frac{1}{\epsilon}\big)$.

This clarifies when and why dimension appears in the complexity.

## B    LOWER BOUNDS FOR EXHAUSTIVE/RANDOM ZEROTH-ORDER SEARCH

We formalize the intrinsic slowness of exhaustive (grid) search and best-of-$N$ random sampling when only function values (or preferences) are used without directional information. The hard instance is the strongly concave quadratic

$$r(z) \;=\; r(z^\star) - \tfrac{\mu}{2} \, \|z - z^\star\|_2^2, \qquad z \in B_R(z^\star) \subset \mathbb{R}^d,$$

whose $\epsilon$-optimal set is the ball $B_{\rho_\epsilon}(z^\star)$ with radius $\rho_\epsilon = \sqrt{2\epsilon/\mu}$.

**Proposition 2** (Grid-search lower bound). *Let $B_R(z^\star) \subset \mathbb{R}^d$ and a hypercubic grid of spacing $h$. Its covering radius is $\rho = \frac{\sqrt{d}\,h}{2}$. To guarantee that* for all *placements of $z^\star$ there exists a grid point in the $\epsilon$-optimal ball $B_{\rho_\epsilon}(z^\star)$ with $\rho_\epsilon = \sqrt{2\epsilon/\mu}$, it suffices that $\rho \le \rho_\epsilon$ (i.e., $h \le 2\rho_\epsilon/\sqrt{d}$). Furthermore, any such grid restricted to $B_R(z^\star)$ must contain at least*

$$N \;\ge\; \left(\frac{R}{\rho}\right)^d \;=\; \left(\frac{R\sqrt{d}}{2\rho_\epsilon}\right)^d \;=\; \left(\frac{\mu R^2 d}{8\,\epsilon}\right)^{d/2}$$

*points. Hence exhaustive grid search is exponential in $d$ and polynomial in $1/\epsilon$ with exponent $d/2$ on this family.*

*Proof.* Coverage of $B_R(z^\star)$ by $N$ balls of radius $\rho$ centered at grid points implies $N V_d \rho^d \ge V_d R^d$, hence $N \ge (R/\rho)^d$. With $\rho = \sqrt{d}\,h/2$ and $h \le 2\rho_\epsilon/\sqrt{d}$, we obtain $N \ge (R\sqrt{d}/(2\rho_\epsilon))^d$. Substitute $\rho_\epsilon = \sqrt{2\epsilon/\mu}$ to conclude. $\qquad\square$

**Proposition 3** (Best-of-$N$ random sampling lower bound). *Draw $X_1, \ldots, X_N \overset{i.i.d.}{\sim} \mathrm{Unif}(B_R(z^\star))$ and let $\hat{z} = \arg\max_i r(X_i)$ for $r(z) = r(z^\star) - \frac{\mu}{2}\|z - z^\star\|_2^2$. Then with $a \triangleq 2/d$,*

$$\mathbb{E}[r(z^\star) - r(\hat{z})] = \frac{\mu R^2}{2}\, N\, \mathrm{B}(1+a, N) = \frac{\mu R^2}{2}\, \Gamma(1+a)\, \frac{\Gamma(N+1)}{\Gamma(N+1+a)}.$$

*Moreover, for all $d \ge 1$ (so $a \in (0, 2]$),*

$$\frac{\Gamma(N+1)}{\Gamma(N+1+a)} \;\ge\; (N+2)^{-a},$$

*and thus*

$$\mathbb{E}[r(z^\star) - r(\hat{z})] \;\ge\; \frac{\mu R^2}{2}\, \Gamma\!\left(1 + \frac{2}{d}\right)(N+2)^{-\frac{2}{d}} \;=\; \Omega\!\left(N^{-\frac{2}{d}}\right).$$

*Proof.* Let $R_i = \|X_i - z^\star\|_2$ and $R_{\min} = \min_i R_i$. The CDF of $R_{\min}$ is $F(r) = 1 - (1 - (r/R)^d)^N$ for $r \in [0, R]$. Differentiating, $f(r) = N d r^{d-1} R^{-d} (1 - (r/R)^d)^{N-1}$. Then

$$\mathbb{E}[R_{\min}^2] = \int_0^R r^2 f(r)\, dr = N R^2 \int_0^1 t^{\frac{2}{d}} (1-t)^{N-1} dt = N R^2\, \mathrm{B}\!\left(1 + \tfrac{2}{d},\, N\right),$$

where $t = (r/R)^d$ and B is the Beta function. Using $\mathrm{B}(a, b) = \frac{\Gamma(a)\Gamma(b)}{\Gamma(a+b)}$ gives the exact expression. For the bound, we use the inequality $\Gamma(N+1)/\Gamma(N+1+a) \ge (N+2)^{-a}$ which holds for all $a \in (0, 2]$ and $N \ge 1$. $\qquad\square$

## C    EXTENDED EXPERIMENT SECTION

### C.1    IMPLEMENTATION DETAILS

**SVG Code Optimization.**    We employ a tournament-style approach where `gpt-5-mini` generates SVG/TikZ code that gets rendered to PNG images for pairwise aesthetic comparisons by a separate instance of the same model acting as judge. The system maintains a "champion" design that only updates when both A-vs-B and B-vs-A orderings consistently agree on a winner, accumulating winning rationales into the generation prompt to guide aesthetic improvements across iterations. The judge provides natural language rationales explaining aesthetic preferences that inform subsequent generations.

| | Method | ADRB1 | PGR | PPARA | PPARG | CDK2 | F2 |
|---|---|---|---|---|---|---|---|
| DOCKSTRING (N=260155) | Top 50% | 5.305 | 3.478 | 4.549 | 4.210 | 4.385 | 4.168 |
| | Top 90% | 8.785 | 7.878 | 7.987 | 7.658 | 7.733 | 7.477 |
| | Top 99% | 9.620 | 8.703 | 8.718 | 8.449 | 8.453 | 8.139 |
| | Top 99.9% | 10.209 | 9.260 | 9.230 | 9.012 | 8.979 | 8.722 |
| | Top 99.99% | 10.742 | 9.723 | 9.821 | 9.518 | 9.509 | 9.252 |
| | Best Molecule | 11.330 | 9.742 | 9.907 | 9.529 | 9.534 | 9.311 |
| | GP-BO[†] | $10.552 \pm 0.140$ | $9.307 \pm 0.177$ | $9.680 \pm 0.337$ | $9.485 \pm 0.279$ | $9.067 \pm 0.289$ | $8.686 \pm 0.068$ |
| | Graph MCTS[†] | $8.883 \pm 0.826$ | $7.819 \pm 0.319$ | $7.363 \pm 0.935$ | $7.134 \pm 0.855$ | $7.777 \pm 0.723$ | $6.310 \pm 0.704$ |
| | Graph GA[†] | $10.249 \pm 1.002$ | $8.793 \pm 0.497$ | $9.211 \pm 0.343$ | $8.769 \pm 0.432$ | $8.652 \pm 0.449$ | $8.900 \pm 0.817$ |
| | SMILES GA | $9.334 \pm 0.237$ | $8.335 \pm 0.276$ | $9.052 \pm 0.484$ | $8.560 \pm 0.346$ | $8.268 \pm 0.170$ | $7.984 \pm 0.554$ |
| | REINVENT | $9.867 \pm 0.522$ | $8.604 \pm 0.483$ | $8.735 \pm 0.120$ | $9.054 \pm 0.153$ | $8.695 \pm 0.370$ | $8.441 \pm 0.535$ |
| | No Feedback (Best-of-N) | $6.190 \pm 0.821$ | $8.619 \pm 0.562$ | $8.230 \pm 0.628$ | $8.633 \pm 0.549$ | $8.300 \pm 0.620$ | $8.793 \pm 0.921$ |
| | Random Feedback | $6.604 \pm 0.577$ | $8.385 \pm 0.258$ | $8.276 \pm 0.628$ | $6.780 \pm 0.523$ | $8.793 \pm 0.921$ | $7.993 \pm 0.663$ |
| | Minimal Feedback | $5.863 \pm 0.428$ | $8.779 \pm 0.633$ | $8.507 \pm 0.428$ | $7.998 \pm 0.571$ | $9.439 \pm 0.922$ | $8.420 \pm 0.315$ |
| | TextGrad | $8.531 \pm 0.278$ | $8.057 \pm 0.383$ | $7.953 \pm 0.160$ | $7.256 \pm 0.886$ | $8.174 \pm 0.395$ | $7.357 \pm 0.821$ |
| | Feedback Descent | $\mathbf{10.623} \pm 0.112$ | $\mathbf{9.615} \pm 0.158$ | $\mathbf{9.919} \pm 0.305$ | $\mathbf{10.187} \pm 0.253$ | $\mathbf{9.803} \pm 0.267$ | $\mathbf{9.300} \pm 0.062$ |

Table 4: Full results for molecule optimization on six protein targets. For each target, the top generative result is in **bold**, and any population in the DOCKSTRING database that exceeds the best generative result is underlined. **Feedback Descent rivals or surpasses specialized molecular optimizers across all six targets.**

**IFBench Prompt Optimization.** We closely follow the setting of Agrawal et al. (2025) for this experiment. We use their two-stage DSPy program with the `gpt-4.1-mini` model and temperature 1.0 for the solver and 0.0 for proposer/tagger to balance exploration and precision. To compare two prompts, we go through the training set to identify examples where program A succeeds and B fails, A fails and B succeeds, both fail, or both succeed, creating four explicit quadrants for analysis. We compute lift and precision/recall metrics on hypothesis tags, where lift measures the base rate of each event and the rate at which it occurs under a subset.

**Molecule Optimization.** We implement molecular optimization using the DOCKSTRING package (García-Ortegón et al., 2022) for protein-ligand docking simulations across six therapeutic targets. The system begins with three simple seed molecules (acetamide, pentane, benzene) and progressively evolves SMILES strings through iterative feedback loops that incorporate RDKit molecular properties, protein binding site information, and similarity comparisons to approved drugs as metadata. We use the combined score function suggested by DOCKSTRING:

$$s_{\text{overall}}(\text{molecule}, \text{protein}) = -\texttt{Vina}(\text{molecule}, \text{protein}) - 10 * (1 - \texttt{QED}(\text{molecule})), \quad (4)$$

where `Vina` provides the binding affinity prediction (kcal/mol, more negative is better) and the QED penalty term penalizes molecules with poor drug-likeness, with lower overall scores indicating better molecules that balance binding strength and drug-like properties. Note that QED scores range from 0 to 1 while Vina scores typically range from $-3.0$ to $-12.0$ kcal/mol. For Feedback Descent, we use a batch size of 8 and top-k selection of 10 examples.

## C.2 ADDITIONAL RESULTS

Fig. 6 shows that across all protein targets, the discovered molecules extend beyond the DOCK-STRING baseline along both axes. The resulting Pareto frontiers illustrate consistent improvements in the joint trade-off between docking affinity and drug-likeness, highlighting that feedback-guided search yields coordinated gains rather than isolated outliers.

Fig. 7 shows optimization trajectories across all six protein targets. In each case, Feedback Descent reaches strong binding scores within the first few hundred oracle calls, while the competing specialized methods often plateau early (e.g., GRAPH-MCTS) or require substantially more evaluations to approach similar performance (e.g., SMILES-GA, GP-BO). Overall, the method is competitive with these baselines and in several cases outperforms them, suggesting that textual feedback provides a broadly effective and robust optimization signal across diverse binding targets.

## C.3 PROMPT TEMPLATES

We use the following prompt for the judge for the Anatomy SVG task. The rubrics for the other tasks are written in a similar style, translating a particular aesthetic into operational rules that minimize ambiguity.

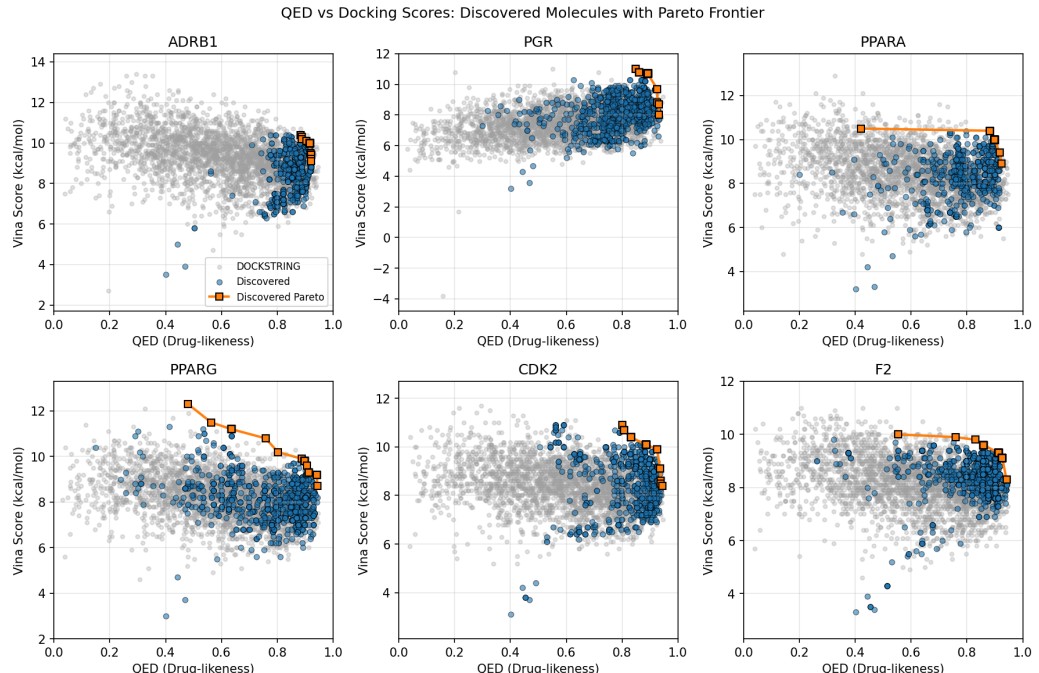

Figure 6: Pareto frontiers of discovered molecules (blue) compared against molecules in the DOCKSTRING dataset (gray) across six protein targets. The highlighted orange markers indicate molecules on the discovered Pareto frontier, achieving joint improvements in docking affinity (Vina score) and drug-likeness (QED).

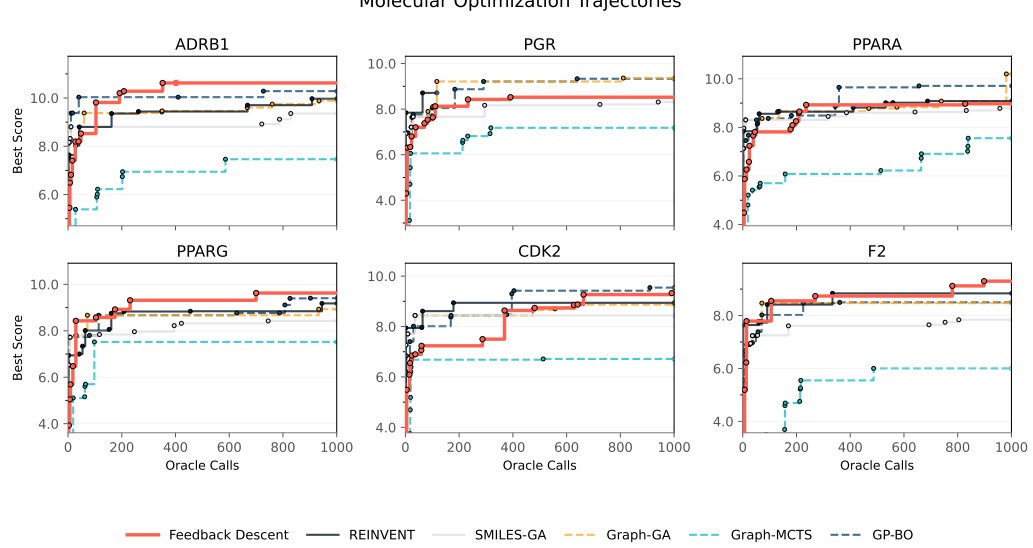

Figure 7: **Optimization trajectories across six protein targets.** Feedback Descent consistently attains higher docking scores with fewer oracle calls compared to standard molecular optimization baselines (REINVENT, SMILES-GA, GRAPH-GA, GRAPH-MCTS, GP-BO).

**Anatomy Judge Rubric**

```
RUBRIC NAME: Anatomical Realism
```

```
INTENT: Believable equine anatomy with a plausible horn; form,
proportion, and structure matter most.

NON-NEGOTIABLES:
- Recognizable equine proportions; head, neck, torso, four legs, mane
, tail, horn present.
- Limbs connect anatomically; joints and hooves indicated.

CRITICAL BENCHMARKS (must evaluate these first):
1. Head-Neck Proportion: Neck length should be ~1.5x head length;
head meets neck high on shoulders
2. Body Square: Body length (shoulder to buttock) ~ height at withers
; chest depth ~ elbow height
3. Leg Structure: Proper joint articulation with elbow under withers;
 fetlock/pastern angles 45-55 deg when standing; all four limbs
distinct and correctly connected

WHAT TO REWARD:
- Correct limb count and articulation; mass distribution that could
stand or move.
- Horn integrates naturally with the skull (frontal bone center, 2-3"
 above eye line).
- Subtle shading or line variation conveying volume.
- Ground contact or cast shadow for grounding.
- Visible muscle definition suggesting tension/relaxation appropriate
 to pose.
- Differentiated hair textures: short coat vs coarse mane/tail
strands.
- Anatomical landmarks: withers prominence, gaskin curve.

WHAT TO PENALIZE:
- Missing or fused legs; impossible joints; balloon torsos.
- Flat cardboard profiles with no sense of volume.
- Decorative effects that obscure structure.
- Disney-fied proportions (oversized eyes, baby-like features).
- Horn placement anywhere except frontal bone center (2-3" above eye
line).

TIEBREAKERS:
- Prefer the image with more accurate limb/neck/head proportions.
- If both are plausible, choose the one with better weight and
grounding.
```

We use the following prompt templates for candidate generation and rationale generation for prompt optimization.

---

**Prompt Template IFBench Candidate Generation**

```
You are tasked with improving an assistant's prompt based on task
data, examples, and feedback.

## Current Prompts
**Approach A (Baseline):**
```python
{prompt_a_dict}
```

**Approach B (Challenger):**
```python
{prompt_b_dict}
```
```

```
## Training Signals
{comparison}

## Step 1: Task Inference
- Read the examples and feedback carefully.
- Infer the underlying task structure, required input/output forms,
and success criteria.
- Identify implicit constraints not explicitly stated in the original
 prompts.

## Step 2: Knowledge Preservation
- Extract and encode domain-specific facts, constraints, and
conventions discovered in the examples.
- Include niche technical details that may not be obvious to a model
without this context.
- Distill general strategies the assistant used successfully.

## Step 3: Failure Analysis
- Identify recurring mistakes and failure modes.
- Devise principle-based instructions to avoid them.
- When both approaches fail on a pattern, invent a new, generalizable
 strategy.

## Step 4: Prompt Synthesis
Write a new instruction prompt that:
1. Captures the task understanding (Step 1).
2. Preserves domain-specific facts (Step 2).
3. Embeds strategies that worked and guards against failures (Step 3)
.
4. Remains SHORT, PRINCIPLE-BASED, and free of training-set
overfitting.
5. Works robustly on UNSEEN TEST DATA.

The prompt must be a Python dictionary with the following keys:
{module_keys_description}

Output EXACTLY in this format:

```python
{prompt_template}
```\"""
You are improving {artifact_type}. Here is the previous feedback:

Iteration 1: {rationale_1}
Iteration 2: {rationale_2}
...
Iteration {n}: {rationale_n}

Current best version:
{current_best}

Generate an improved version that addresses the feedback while
preserving existing strengths. Focus on the most actionable
suggestions from the rationales above.
```

**Prompt Template for Molecule Optimization**

```
You are a helpful assistant that proposes a single SMILES string
corresponding to a small molecule, based on the task explanation and
information about previous molecules that have been proposed. Your
```

```
proposed molecule should score highly on the {benchmark_name}
benchmark.

Task explanation: Maximize -{benchmark_name} - 10 * (1 - QED): larger
 values mean stronger {benchmark_name} binding and high drug-likeness
.
{protein_info_xml}

Your output format should be EXACTLY as follows:

<reasoning>[Chemical hypothesis linking structural changes to
expected binding/QED improvements]</reasoning>
<smiles>[Single SMILES string]</smiles>

Generate a small molecule based on these previous examples (sorted in
 descending order of score):

{examples_text}
```

### Example of Protein Metadata (ADRB1)

```
{ 'target': 'ADRB1', 'accession': 'P08588', 'regions': { '
transmembrane': [[56, 84], [94, 120], [133, 154], [173, 196], [223,
248], [320, 349], [355, 377]], 'extracellular': [[1, 55], [121, 132],
 [197, 222], [350, 354]], 'cytoplasmic': [[85, 93], [155, 172], [249,
 319], [378, 477]], 'disordered': [[269, 307], [403, 477]]}, '
critical_residues': {'mutagenesis': [{'position': [474, 474], '
description': 'Loss of interaction with GOPC.'}, {'position': [474,
474], 'description': 'Loss of interaction with GOPC; when associated
with A-477.'}, {'position': [475, 475], 'description': 'Loss of
interaction with GOPC. Loss of interaction with RAPGEF2. Abolishes
agonist-induced Ras activation.'}, {'position': [475, 475], '
description': 'Loss of interaction with RAPGEF2.'}, {'position':
[475, 475], 'description': 'Partial loss of interaction with GOPC.'},
 {'position': [476, 476], 'description': 'Partial loss of interaction
 with GOPC.'}, {'position': [477, 477], 'description': 'Loss of
interaction with GOPC.'}, {'position': [477, 477], 'description': '
Loss of interaction with RAPGEF2. Abolishes agonist-induced Ras
activation.'}], 'natural_variants': [{'position': [26, 26], '
description': 'in dbSNP:rs34844626'}, {'position': [29, 29], '
description': 'in dbSNP:rs35720093'}, {'position': [31, 31], '
description': 'in dbSNP:rs35230616'}, {'position': [49, 49], '
description': 'correlated with low mean resting heart rate and
decreased mortality risk in patients with congestive heart failure;
dbSNP:rs1801252'}, {'position': [187, 187], 'description': 'found in
individuals with short sleep; results in decreased adenylate cyclase-
activating adrenergic receptor signaling; decreased protein stability
; dbSNP:rs776439595'}, {'position': [389, 389], 'description': '
increased beta1-adrenergic receptor activity; increased basal
activity and increased coupling to heterotrimeric G protein Gs that
stimulates the adenylyl cyclase; dbSNP:rs1801253'}, {'position':
[399, 399], 'description': 'in dbSNP:rs36052953'}, {'position': [405,
 405], 'description': 'in dbSNP:rs35705839'}]}}
```

### Example of Molecule Metadata (CCCCC)

```
valid: 'True'
score: '-1.9121449019886678'
metadata:
```

```
CanonicalSMILES: CCCCC
InChIKey: OFBQJSOFQDEBGM-UHFFFAOYSA-N
MolecularFormula: C5H12
ExactMass: '72.093900384'
FormalCharge: '0'
AtomCount: '5'
HeavyAtomCount: '5'
HeteroAtomCount: '0'
BondCount: '4'
Sp3CarbonFraction: '1.0'
RingCount: '0'
AromaticRingCount: '0'
AliphaticRingCount: '0'
RotatableBondCount: '2'
StereoCenterCount: '0'
MurckoScaffold: ''
LogP: '2.1965000000000003'
TopologicalPolarSurfaceArea: '0.0'
MolarRefractivity: '25.19899999999999'
HBondDonorCount: '0'
HBondAcceptorCount: '0'
BertzComplexityIndex: '7.5097750043269365'
BalabanJIndex: 2.19060968716425
HallKierAlpha: '0.0'
Kappa1: '5.0'
Chi0v: '4.121320343559642'
TotalEState: 8.5
MinEState: 1.34375
MaxEState: 2.2118055555555554
PEOE_VSA6: '33.10993926815928'
SlogP_VSA5: '33.10993926815928'
BCUTp_1h: '13.744962415414642'
AccessibleSurfaceArea: '34.19901948541599'
FunctionalGroups: []
StructuralAlerts: []
QuantitativeDrugLikeness: '0.4687855098011332'
SyntheticAccessibility: '1.699621281696647'
NaturalProductLikeness: '0.09749981667944'
```

## C.4 DISCOVERED PROMPTS FOR IFBENCH

Below, we show the discovered prompts for Qwen3-8B and GPT-4.1-mini.

**ensure_correct_response_module, Qwen3-8B (acc=44.22)**

```
Extract every explicit constraint: order/sequence (e.g., repeat
verbatim first, nothing before it; required exact ending), verbatim
text/keywords (case, spacing, punctuation), forbidden items, numeric/
format limits (exact/min sentences, words, characters; counts of
letters/words/capitalized words; number/format of bullets/items),
tone/style, math/logic (units, rounding), and formatting bans (e.g.,
no code blocks). Build a checklist. Validate the draft: (1) If repeat
-first or 'nothing before' is required, ensure the very first
character starts the repeated text; copy it exactly; no quotes/
headers/spaces/blank lines before it. (2) If an exact ending is
required, the final characters are exactly that phrase with nothing
after. (3) All required phrases/keywords included as specified (
respect case/order if stated). (4) Numeric/format limits match
precisely, including sentence count and capitalized-word count;
control with short, simple sentences and standard punctuation; avoid
abbreviations/ellipses/decimals that can alter sentence counts unless
```

```
 necessary. (5) Math is correct; apply requested rounding/units. (6)
Tone met; no forbidden items. If constraints conflict, prioritize:
order/sequence > verbatim/ending > forbidden items > numeric/format
limits (incl. sentence and capitalized-word counts) > keywords/tone >
 any extras. Fix issues and re-check. Remove trailing spaces/newlines
.",
```

**generate_response_module, Qwen3-8B (acc=44.22)**

```
Read the prompt and list constraints: sequence (repeat verbatim first
; nothing before; required exact ending), scope of counts (entire
response vs answer only), exact phrases/keywords (case/order),
forbidden items, numeric limits (exact/min sentences, words,
characters; occurrences; number/format of bullets/items; count of
capitalized words), tone, and any math/logic with units/rounding.
Plan the structure accordingly. If required to repeat the request
verbatim at the beginning, copy it exactly and place it first with
nothing before; do not add quotes; then proceed to the answer (use a
single newline as a separator only if not forbidden). Scope all
counts as specified; if unspecified, apply them to the entire
response. Meet numeric limits exactly: control sentence count with
simple sentences and standard punctuation; avoid abbreviations/
ellipses/parentheticals; deliberately include the needed number of
Capitalized words and count them. Include required keywords/phrases
in the stated order/case; exclude forbidden items. Do computations
accurately; follow rounding/units. If a specific ending is required,
ensure your final characters are exactly that phrase. Provide step-by
-step explanation only if explicitly requested; otherwise be concise.
 Before finalizing, recount/recheck against the constraint list and
adjust. Remove trailing whitespace.
```

**generate_response_module, GPT-4.1-mini (acc=54.59)**

```
Pre-check for compliance and correctness: 1) Parse the task into a
compact internal spec: goal and success criteria; exact required
outputs; structure (counts/order/labels/delimiters); required first/
last tokens; exact literals to reproduce and their placement (
preserve casing/spacing/punctuation); content rules (required/
forbidden items and exact occurrence/length limits); language/
modality; numeric rules (use only provided data; units/conversions;
round only at the end); safety/policy limits. 2) Apply instruction
hierarchy (system > developer > user); resolve by specificity and
recency. If full compliance is impossible or unsafe, produce the
smallest safe compliant output; do not invent facts. 3) Numbers:
extract data and units, normalize units, compute precisely, verify
totals/consistency, delay rounding. 4) Verbatim/echo: copy literals
exactly, respect stated inclusion/exclusion boundaries, no
normalization or padding before/after echoed segments. 5) Final audit
: confirm structure and counts, required positions and boundary
tokens, verbatim exactness, occurrence/length limits, absence of
forbidden items, language/modality lock, numeric units and rounding,
safety compliance, and no extra text.
```

**ensure_correct_response_module, GPT-4.1-mini (acc=**54.59**)**

```
Plan then write: decide the exact output shape from the spec (
sections/items/order/labels/delimiters) and fix boundary tokens and
literal placements. Lock the requested language and modality. Use
only provided data for any calculations; normalize units; apply
rounding at the end. Draft the smallest content that satisfies all
constraints; enforce required/forbidden items and exact occurrence/
length counts while writing. Self-check and repair: recount structure
 and counts; verify first/last tokens and required positions; ensure
verbatim correctness with no added/omitted characters or padding;
confirm numeric correctness and units; ensure safety/policy
compliance. Output only the final compliant answer.
```

