# OpenReview forum: "Feedback Descent: Open-Ended Text Optimization via Pairwise Comparison"
_ICLR.cc/2026/Conference — ICLR 2026 Conference Desk Rejected Submission_

### Official Review · Reviewer_ckan · 2025-10-30

**Soundness:** 3
**Presentation:** 3
**Contribution:** 3
**Rating:** 8
**Confidence:** 4

**Summary:**

The authors propose a very interesting idea of optimizing in the text space via explicit feedback instead optimizing in the latent space. They use pairwise preferences with text to provide a direction and show that even with imperfect correlation of the feedback and the gradients, the aggregated messages across failures guides the model towards improving the result. They evaluate the model on three domains and show in SVG design iterative feedback produces improvements above and beyond direct prompting from both scratch and rubric-aware initializations. On the molecule discovery task the proposed approach outperforms RL based approaches. Overall this is a very interesting area of research direction with good results as a start.

**Strengths:**

The authors identify an important area of research and apply it to three different domains where they show the performance of the proposed approach is competitive. Exploiting the text space to provide feedback is an interesting idea. It can be thought of as analogous to gradient descent when the direction of text feedback aligns with the direction of gradients. Even if the latest feedback is not useful as the authors show the accumulation of past feedback and why it failed helps the model to refine the output.
The first proposition shows that how the proposed approach is advantageous given it does not depend on the hidden dimension unlike the other methods. Both propositions mainly highlight the dimensionality as a key bottleneck.

Results on the SVG designs look impressive with the proposed approach beating baselines in most cases. In the docking example the proposed approach augments the LLM with feedback on the key properties related to the task.

**Weaknesses:**

As the authors mention in the limitations section, the model will rely on strong evaluators which can be challenging in certain domains for e.g. in designing antibodies. If there is a way to combine multiple feedbacks such as use an ensemble of feedback from text and latent space and then guide the model it might make the approach robust. And second strictly following either feedback might also be limiting and there is definitely scope to be creative on "how to use the feedback".

**Questions:**

In the equation for the proposition 1 when the algorithm calculate the \delta(r) for each rationale, how does it determine the positively aligned past rationales? Is there another module to compare whether the model refined the feedback directly based on the current rationale? What defines a "useless rationale"? Can it be that the proposed direction is not completely off but would take longer towards the refinement and if so is it considered a bad rationale?

For the second proposition, it has been shown that DPO based approaches can extrapolate to spaces (and hence for e.g. propose new outputs) that have better fitness say e.g. in the protein domain. Can such feedback be used in conjunction with DPO to reach to the better designs faster?

---

> ### Author Response · Authors · 2025-11-28
> **rebuttal (1/1)**
>
> Thank you for your thoughtful feedback. We address specific concerns below.
>
> > What defines a "useless rationale"?
>
> We provide an empirical definition of "useful rationale" and back it with a new experiment. We can view a rationale as "useful" if the generator can act on it. To measure this, we present an LLM judge (gpt-4o-mini) with: (1) a newly generated molecule, (2) the actual feedback text with correct property metadata, and (3) a scrambled control where property metadata is randomly shuffled between molecules. The judge is asked to determine which feedback set the molecule more closely follows. We expect actual feedback to win >50% if the rationale is useful under this definition, and ~50% otherwise (the null hypothesis). Across 400 comparisons over 4 protein targets (ADRB1, CDK2, F2, PPARG), the true feedback wins 81% of head-to-head comparisons (binomial test $p < 10^{-30}$):
>
> |          | ADRB1 | CDK2 | F2  | PPARG | Aggregate |
> | -------- | ----- | ---- | --- | ----- | --------- |
> | Win Rate | 76%   | 86%  | 79% | 83%   | 81%       |
>
> This overwhelmingly rejects the null hypothesis and demonstrates that the rationales are "useful" under this definition.
>
> > If there is a way to combine multiple feedbacks such as use an ensemble of feedback from text and latent space and then guide the model it might make the approach robust..strictly following either feedback might also be limiting and there is definitely scope to be creative on "how to use the feedback".
>
> We agree that this is a promising direction. Jointly optimizing via text-space feedback and weight-space updates (e.g., DPO, RLHF) could combine the complementary strengths of both approaches: text feedback provides interpretable, high-bandwidth guidance for discrete artifacts, while weight updates enable continuous refinement of the generator's policy. We leave this hybrid approach to future work.
>
> > In the equation for the proposition 1 when the algorithm calculate the delta(r) for each rationale, how does it determine the positively aligned past rationales? Is there another module to compare whether the model refined the feedback directly based on the current rationale?
>
> Proposition 1 assumes that alignment is high. In practice, the LLM implicitly integrates all accumulated feedback when generating the next candidate. We will clarify that Proposition 1 provides intuition for why accumulating feedback helps, not a prescription for how to implement it.
>
> > For the second proposition, it has been shown that DPO based approaches can extrapolate to spaces (and hence for e.g. propose new outputs) that have better fitness say e.g. in the protein domain. Can such feedback be used in conjunction with DPO to reach to the better designs faster?
>
> Extrapolation and generalization are properties of any learning algorithm, including DPO and our method. Our experiments show that Feedback Descent exhibits "extrapolation" in a quite wide sense, i.e., it discovers novel molecules (Tversky similarity < 1.0 to any known molecule) that outperform the best molecule in the entire 260k DOCKSTRING database on multiple targets. We agree that combining weight-space and text-space learning is an interesting direction.

---

### Official Review · Reviewer_jREt · 2025-10-31

**Soundness:** 2
**Presentation:** 2
**Contribution:** 1
**Rating:** 2
**Confidence:** 3

**Summary:**

This paper presents Feedback Descent, a general framework for inference-time optimization of text-based artifacts. Instead of learning from scalar rewards or binary preferences, it leverages pairwise comparisons augmented with textual rationales. In each iteration, a candidate is generated using prior feedback, compared to the current best, and either accepted or rejected based on preference and rationale. This loop produces directional edits in semantic space, acting like approximate gradients. The method is tested on three tasks: SVG design, prompt optimization, and molecular discovery.

**Strengths:**

The method addresses a real limitation in preference learning by incorporating rich feedback rather than collapsing supervision into binary signals. It is model-agnostic and requires no parameter updates, making it widely applicable. Experiments span distinct domains and show  improvements over well-established baselines. The use of textual rationales to guide generation is intuitive and aligns with emerging trends in LLM usage. The SVG and molecular tasks show the advantages of iterative feedback, even over strong initial prompts. The theoretical section provides useful insight, framing textual feedback as noisy but directionally useful signals that improve sample efficiency in high-dimensional tasks.

**Weaknesses:**

1. The framework heavily depends on high-quality evaluators that can provide meaningful and consistent textual rationales. If the feedback is noisy, vague, or inconsistent, the system may stagnate or regress.

2. The update step only keeps one best candidate per iteration, which may limit diversity and exploration.

3. The paper lacks ablation studies showing how performance changes when textual feedback is removed or corrupted.

4. although the method is said to be domain-general, all tasks still rely on well-defined evaluators or rubrics. In domains lacking reliable feedback sources, its applicability remains unclear.

**Questions:**

1. How does the method perform when the textual feedback is noisy, irrelevant, or inconsistent? Would it still improve over time, or does it degrade quickly?

2. Why did the authors choose to reset the feedback history after each accepted update? Could cumulative memory of feedback help guide long-term edits better?

3. Have the authors tested multi-candidate selection instead of greedy updates?

4. In creative tasks without clear evaluation rubrics, can this approach still function reliably, or does it require strong evaluators to be useful at all?

---

> ### Author Response · Authors · 2025-11-28
> **rebuttal (1/1)**
>
> Thank you for your thoughtful feedback. We address specific concerns below.
>
> > The framework heavily depends on high-quality evaluators that can provide meaningful and consistent textual rationales. If the feedback is noisy, vague, or inconsistent, the system may stagnate or regress.
> > How does the method perform when the textual feedback is noisy, irrelevant, or inconsistent? Would it still improve over time, or does it degrade quickly?
>
> We tested corrupted feedback by randomly shuffling rationales between molecule pairs at 25%, 50%, and 100% noise levels:
>
> | Benchmark | No Noise  | 25% Noise | 50% Noise | 100% Noise |
> | --------- | --------- | --------- | --------- | ---------- |
> | ADRB1     | **10.62** | 9.28      | 10.21     | 8.62       |
> | PGR       | **9.62**  | 9.14      | 8.92      | 6.05       |
> | PPARG     | **10.19** | 8.16      | 8.75      | 8.22       |
>
> The method degrades gracefully: 50% noise still achieves scores above the 99.9th percentile of DOCKSTRING for ADRB1 and PGR, and above the 99th percentile for PPARG. Even 100% noise (fully random feedback) outperforms Best-of-N sampling on most targets, suggesting the iterative structure itself provides some value beyond the feedback content. However, performance clearly improves with feedback quality, validating that the generator uses rationale information when available.
>
> > The update step only keeps one best candidate per iteration, which may limit diversity and exploration.
> > Have the authors tested multi-candidate selection instead of greedy updates?
>
> We have not tested multi-candidate selection, but note that our batch size (5 candidates per iteration) already provides within-iteration diversity. The greedy update rule selects the best among these candidates, so exploration occurs through batch sampling rather than population maintenance. Maintaining a Pareto frontier or diverse population across iterations is an interesting extension; we expect it would help in multi-objective settings at the cost of additional complexity.
>
> > Why did the authors choose to reset the feedback history after each accepted update? Could cumulative memory of feedback help guide long-term edits better?
>
> We reset feedback after accepted updates for two reasons: (1) **context limits**: accumulating all feedback across hundreds of iterations would exceed model context windows; (2) **relevance decay**: feedback about why molecule A was worse than molecule B becomes less actionable once we've moved to molecule C. After an accepted update, the new best candidate renders prior rejection rationales obsolete. However, we retain feedback across _rejected_ iterations, which accumulates information about which changes fail relative to the current best. This asymmetric memory preserves useful negative examples while avoiding confusion from outdated comparisons.
>
> > In creative tasks without clear evaluation rubrics, can this approach still function reliably, or does it require strong evaluators to be useful at all?
>
> Our approach fundamentally requires evaluators that can distinguish better from worse outputs. In domains without ground truth or consensus (e.g., creative writing, art), the method's utility depends on whether the evaluator's preferences align with the user's goals. This is not unique to our method: all optimization requires an objective function. We position Feedback Descent as a tool for settings where evaluation is available (e.g., automated metrics, expert judges, or LLM proxies for human preferences). We will clarify this scope in the introduction and limitations.

---

### Official Review · Reviewer_GXdf · 2025-11-01

**Soundness:** 2
**Presentation:** 3
**Contribution:** 2
**Rating:** 4
**Confidence:** 3

**Summary:**

This paper proposes **Feedback Descent**, an inference-time optimization framework that iteratively improves text-based artifacts (SVG code, prompts, molecules) through pairwise comparisons augmented with textual rationales. Unlike standard preference learning methods that compress judgments into scalars or binary signals, Feedback Descent treats textual feedback as directional information analogous to gradients in continuous optimization.

The authors validate their approach across three diverse domains: SVG aesthetic optimization, prompt engineering on IFBench, and molecule discovery on DOCKSTRING, demonstrating competitive or superior performance compared to state-of-the-art baselines including GEPA, GRPO, REINVENT, and graph-based molecular optimizers.

The authors are refreshingly honest that textual feedback is not a literal gradient, instead framing it as a "heuristic directional cue" that provides higher-bandwidth supervision than scalar rewards or binary preferences. In this sense, Feedback Descent extends the TextGrad philosophy by emphasizing pairwise comparisons with accumulated feedback rather than self-reflection alone.

**Strengths:**

1. The paper is generally well-written with clear motivation and contributions.

2. Algorithm 1 is simple, self-contained, and reproducible.

3. Inference-time optimization with no weight updates is valuable for practitioners. (Minor note: "SVG" should be expanded to Scalable Vector Graphics at first mention for accessibility.)

4. The framework is validated on three qualitatively different tasks:

    - Visual design (SVG),

    - Natural language (prompts), and

    - Chemistry (molecules).

5. The molecule discovery results are particularly strong.

6. It has fair comparisons with matched budgets using comprehensive baselines, including recent methods (GEPA, GRPO).

**Weaknesses:**

1. The authors claim that "a paragraph of feedback contains more Shannon information than a single scalar or bit," which is trivially true in terms of raw bits. However, information content does not equal actionable directional information. Several critical questions remain unaddressed.

2. **Quantitative validation missing:**
Can the authors quantify whether textual feedback actually provides gradient-aligned directions? For example, measure the correlation between feedback-suggested changes and actual objective improvement.

3. **Ablations needed:**
If feedback is noisy, vague, or contradictory across iterations, scalar signals + random search might be more reliable. The paper needs ablations showing comparison to "random descent" (same loop structure but random mutations instead of feedback-guided ones). Maybe adding analysis of when feedback helps vs. when it misleads.

4. **Theory–practice gap:**
The theoretical contribution is undermined by missing formalism. The paper assumes that if feedback provides gradient-like directions, then fast convergence follows — but there’s no natural notion of $z_t + \eta v_t$ for SMILES strings or SVG code. The LLM generates a new $x_{t+1}$ by interpreting textual feedback — this is not vector addition. Also, the mapping $\Phi: S \to Z$ (text to latent space) is never defined or validated.

**Questions:**

1. Can you provide empirical evidence that feedback directions are aligned with improvement directions?

2. Can you provide error bars (mean $\pm$ std) over multiple random seeds for all experiments?

3. Can you run TextGrad on at least one shared task (e.g., molecules) for direct comparison?

4. What is the computational cost (wall-clock time, API calls, dollars) compared to baselines? Are the gains worth the added evaluation expense?

---

> ### Author Response · Authors · 2025-11-28
> **rebuttal (1/2)**
>
> Thank you for your thoughtful feedback. We address specific concerns below.
>
> > Can you run TextGrad on at least one shared task (e.g., molecules) for direct comparison?
>
> Thank you for this suggestion; we agree that this is an important point of comparison. We ran TextGrad for up to 1000 iterations on all 6 protein targets for the DOCKSTRING task.
>
> | Target | TextGrad    | Feedback Descent |
> | ------ | ----------- | ---------------- |
> | ADRB1  | 8.53 ± 0.28 | **10.62 ± 0.11** |
> | PGR    | 8.06 ± 0.38 | **9.62 ± 0.16**  |
> | PPARA  | 7.95 ± 0.16 | **9.92 ± 0.31**  |
> | PPARG  | 7.26 ± 0.89 | **10.19 ± 0.25** |
> | CDK2   | 8.17 ± 0.40 | **9.80 ± 0.27**  |
> | F2     | 7.36 ± 0.82 | **9.30 ± 0.06**  |
>
> Feedback Descent outperforms TextGrad by 1.5–3.0 docking score units across all targets. We attribute this gap to the fact that TextGrad relies on _pointwise_ feedback, where the edit is only based on the scores of the latest molecule, whereas Feedback Descent accumulates feedback across iterations, building context about what structural changes improve binding affinity.
>
> > Can the authors quantify whether textual feedback actually provides gradient-aligned directions?
>
> Thank you for this suggestion, we agree that verifying that the generator's output is actually reflecting the feedback is critical. We conducted an additional experiment to evaluate whether generated molecules are consistent with the feedback shown during optimization. For each iteration, we present an LLM judge (gpt-4o-mini) with: (1) a newly generated molecule, (2) the actual feedback text with correct property metadata, and (3) a scrambled control where property metadata is randomly shuffled between molecules. The judge is asked to determine which feedback set the molecule more closely follows. We expect actual feedback to win >50% if the generator uses property information, and ~50% under the null hypothesis that it ignores feedback. Across 400 comparisons over 4 protein targets (ADRB1, CDK2, F2, PPARG), the true feedback wins 81% of head-to-head comparisons (binomial test $p < 10^{-30}$):
>
> |          | ADRB1 | CDK2 | F2  | PPARG | Aggregate |
> | -------- | ----- | ---- | --- | ----- | --------- |
> | Win Rate | 76%   | 86%  | 79% | 83%   | 81%       |
>
> This overwhelmingly rejects the null hypothesis and demonstrates that the generator reliably reads and incorporates structured property feedback.
>
> > If feedback is noisy, vague, or contradictory across iterations, scalar signals \+ random search might be more reliable. The paper needs ablations showing comparison to "random descent" (same loop structure but random mutations instead of feedback-guided ones). Maybe adding analysis of when feedback helps vs. when it misleads.
>
> We added a "Random Feedback" ablation that uses the same iterative loop but shuffles rationales between molecule pairs, breaking the correspondence between feedback content and actual molecular properties. Results (averaged across 6 targets):
>
> | Method                         | Avg. Docking Score |
> | ------------------------------ | ------------------ |
> | Random Feedback                | 7.81               |
> | Minimal Feedback (binary only) | 8.17               |
> | **Feedback Descent**           | **9.91**           |
>
> Random feedback underperforms even binary-only feedback, confirming that the method relies on feedback content rather than just the iterative structure. The 2.1-unit gap between random and full feedback demonstrates that accurate rationales provide substantial optimization signal.
>
> > The theoretical contribution is undermined by missing formalism...
>
> We acknowledge this gap. The theory provides intuition, not formal guarantees. The LLM's interpretation of feedback is not vector addition, but a learned text-to-text transformation that we hypothesize approximates directional updates in semantic space. We will revise to make this explicit and avoid suggesting a formal equivalence. The mapping is implicitly defined by the LLM's embedding space, but we agree that this warrants empirical validation (as addressed in our correlation analysis above). Additionally, we have revised the paper; now the detailed propositions only appear in the appendix, and we have added a paragraph to the method section laying out the intuition.
>
> > Can you provide error bars (mean ± std) over multiple random seeds for all experiments?
>
> We have now added these for all three experiments in the revised paper, please see the main text tables for the SVG and prompt optimization experiments, and the new extended table in the appendix for the molecule optimization experiment. For the SVG and molecule experiments, we ran additional runs to get 3 seeds for each setting.

---

> > ### Author Response · Authors · 2025-11-28
> > **rebuttal (2/2)**
> >
> > > The authors claim that "a paragraph of feedback contains more Shannon information than a single scalar or bit," which is trivially true in terms of raw bits. However, information content does not equal actionable directional information.
> >
> > We agree that raw bits do not equal actionable information. Our claim is empirical, not information-theoretic: textual feedback enables faster optimization than binary feedback in practice. The evidence:
> >
> > 1. **Ablation gap**: Feedback Descent (9.91 avg) vs. Minimal Feedback with binary-only signal (8.17 avg) — a 1.74 unit improvement from adding rationale text
> > 2. **Feedback alignment**: 81% of generated molecules align with true feedback over scrambled controls, demonstrating that the generator extracts and acts on rationale content
> > 3. **Random feedback degrades performance**: Shuffled rationales (7.81) underperform binary-only (8.17), showing the method relies on feedback _content_, not just feedback _presence_
> >
> > It's unclear how to directly measure the "actionable directional information" in a formal sense, but the optimization benefits are measurable. If the reviewer has suggestions for further analysis, we would be happy to add it to the paper.
> >
> > > What is the computational cost (wall-clock time, API calls, dollars) compared to baselines? Are the gains worth the added evaluation expense?
> >
> > Most of our costs were in API calls. DOCKSTRING is our most expensive experiment. Each optimization run generates 1,000 molecules over 200 iterations (batch size 5), completing in approximately 1.9 hours of wall-clock time on average using GPT-4o-mini. Much of this time is spent waiting for responses from the OpenAI API. The total cost per run is approximately 0.15 USD (200 API calls at \~3,000 input and 500 output tokens each). Across 6 benchmarks with 3 seeds each (18 total runs for 2,000 molecules), total compute time was 68.8 hours, and API costs were \~ 5.40 USD. While iterations are inherently sequential due to the feedback loop, we parallelize within-iteration batch generation and objective evaluation. Additionally, independent runs across different targets and seeds can be fully parallelized: all 18 runs could execute simultaneously, reducing the total wall-clock time to \~3.8 hours.

---

### Official Review · Reviewer_MZQt · 2025-11-03

**Soundness:** 2
**Presentation:** 3
**Contribution:** 1
**Rating:** 2
**Confidence:** 4

**Summary:**

This paper describes a simple LLM-based evolutionary algorithm for optimization, where each comparison within the population is accompanied by a textual description of why it was better. They evaluate on a diverse set of tasks.

**Strengths:**

- The paper correctly identifies that text information contains a lot more information than binary preferences
- Good choice of experiments and baselines, method seems effective
- Writing generally clear and figures look nice!

**Weaknesses:**

- **Scientific question incoherent / inconsistent**: in §1-§4, the paper sets up the scientific question of the paper as "does textual feedback provide a stronger learning signal than binary feedback". Then, §5 seems to answer the question "is direct feedback optimization an effective way to accomplish these tasks". _This is not the original question!!_ In my opinion, the "missing" baseline in each case is the same LLM-based optimizer with the same binary feedback, but _without_ the explanatory text. As far as I can tell, the actual baseline used is a _non-iterative_ (single shot) optimizer, which is quite distinct.
- **"Mathiness" in §3**: §3 felt like it should be entirely omitted and replaced with the sentence "prior work has shown that first-order optimizers are able to explore high-dimensional spaces more effectively than zeroth-order optimizers [citations]". I think introducing a bunch of theorems which never actually get used just makes the paper harder to read without really adding anything.

  The actual theorems themselves seem vacuous: the assumptions for proposition 1 are clearly constructed in a way that omits dependence on the dimensionality (through the assumption of $\mu$-PL), which guarantees that the desired conclusion will hold. In practical problems I imagine the value of $\mu$ in the assumption would depend on d.
- **Relationship to evolutionary algorithms**: the method proposed, which keeps a best candidate, proposes a variation, and keeps the variation if it improves, is an instance of an evolutionary algorithm (aka genetic algorithm). This is a well-established optimization approach so there is nothing wrong with proposing an EA, but I think this should be recognized and acknowledged in the paper.
- **Textual feedback is not a novel insight**: the motivation for the method in lines 30-47 seems to imply that the richness of textual feedback is a novel insight. I'm not sure whether the authors intended this implication, but regardless I want to emphasis that this is _not_ a novel insight. Many people have thought about this, and the reason it is not done more is that it is in fact much more expensive to give text feedback than binary feedback (binary could be <1s, text is probably 10x as long to type out).

**Questions:**

- It was unclear in the paper who the judges are. I presume in all cases the feedback was provided by LLMs, is that correct? (with the exception of the docking scores from vina in dockstring)
- I had some questions about the molecule design task:
  1. As far as I can tell, the feedback from the judge is a bunch of molecular descriptors, so it doesn't actually explain _why_ one molecule was better than another, is that correct?
  2. I did not understand the analysis of figure 3 in lines 461-466. How does the _correlation_ between score and similarity show whether the algorithm is recycling ideas from known molecules? Isn't it sufficient for there to be a _single_ non-novel molecule?

---

> ### Author Response · Authors · 2025-11-28
> **rebuttal (1/2)**
>
> Thank you for your thoughtful feedback. We address specific concerns below.
>
> > The 'missing' baseline in each case is the same LLM-based optimizer with the same binary feedback, but without the explanatory text. The actual baseline used is a non-iterative (single shot) optimizer, which is quite distinct.
>
> We clarify: most of the baselines in our initial submission were iterative (GEPA runs multiple rounds, GRPO trains for multiple epochs, REINVENT runs evolutionary search). However, we agree that comparing to binary-only feedback is an important baseline. We ran three ablations that isolate the contribution of textual rationales:
>
> | Ablation                | Description                               | Avg. Score (6 targets) |
> | ----------------------- | ----------------------------------------- | ---------------------- |
> | No Feedback (Best-of-N) | Sample 1000 molecules, return best        | 8.13                   |
> | Random Feedback         | Same algorithm, shuffled rationales       | 7.81                   |
> | Minimal Feedback        | Binary preference only, no rationale text | 8.17                   |
> | **Feedback Descent**    | Full method                               | **9.91**               |
>
> The gap between Minimal Feedback and Feedback Descent (1.74 docking score units) demonstrates that textual rationales provide substantial value beyond the binary preference signal and iterative structure.
>
> > **Textual feedback is not a novel insight:** The motivation for the method in lines 30-47 seems to imply that the richness of textual feedback is a novel insight. This is not a novel insight.
>
> We cite and discuss multiple prior works that utilize textual feedback (e.g., Constitutional AI, TextGrad). Our contribution is showing that (1) this approach works across diverse domains (SVG, prompts, molecules) and (2) scales to long optimization trajectories where accumulated feedback guides exploration.
>
> > the reason (textual feedback) is not done more is that it is in fact much more expensive to give text feedback than binary feedback (binary could be \<1s, text is probably 10x as long to type out).
>
> We respectfully disagree. When judges are LLMs or automated evaluators (as in our experiments), generating textual feedback adds minimal cost compared to the forward pass itself. In DOCKSTRING, computing docking scores dominates cost; generating descriptors from those scores is negligible. Additionally, for settings with long trajectories, the marginal cost of textual feedback is small: evaluators must process the full trajectory regardless, so generating rationales alongside preferences requires no additional input tokens.
>
> > The method proposed...is an instance of an evolutionary algorithm...this should be recognized and acknowledged in the paper.
>
> In the context of evolutionary algorithms, our contribution is in operationalizing an effective directional mutation operator based on textual feedback. We have revised our related work section to explicitly position our work within the EA literature, citing foundational work on (1+1)-ES and mutation operators, as well as recent work on LLM-guided evolutionary search.
>
> > Section 3 felt like it should be entirely omitted...introducing a bunch of theorems which never actually get used just makes the paper harder to read…The actual theorems themselves seem vacuous.
>
> Thank you for the suggestion. We have moved the formal propositions to the appendix and replaced Section 3 with a concise intuition paragraph in the methods section. We acknowledge that this is a heuristic argument rather than a formal guarantee, and have revised the framing accordingly. We welcome further suggestions on the paper's organization.
>
> > It was unclear in the paper who the judges are. I presume in all cases the feedback was provided by LLMs, is that correct? (with the exception of the docking scores from vina in dockstring)
>
> Correct. We will clarify in each experiment section:
>
> - **SVG optimization**: LLM judge (GPT-4o) evaluates aesthetic quality and provides rationales
> - **Prompt optimization**: Automated benchmark feedback per datapoint, aggregated by LLM due to context limits. The benchmark provides ground-truth correctness and pointwise feedback; the LLM summarizes patterns across datapoints
> - **DOCKSTRING**: Docking score (AutoDock Vina) + molecular descriptors + nearest neighbor matches from known molecule database
>
> > As far as I can tell, the feedback from the judge is a bunch of molecular descriptors, so it doesn't actually explain why one molecule was better than another, is that correct?
>
> Yes, this is correct. The feedback does not literally state "molecule A is better because property X improved," but provides both molecules' descriptors, allowing the generator to infer which properties to adjust. This design choice reflects the fact that domain-specific explanations require chemistry expertise, which we did not want to hard-code. The LLM interprets the structured data to guide edits.

---

> > ### Author Response · Authors · 2025-11-28
> > **rebuttal (2/2)**
> >
> > > I did not understand the analysis of figure 3 in lines 461-466. How does the correlation between score and similarity show whether the algorithm is recycling ideas from known molecules? Isn't it sufficient for there to be a single non-novel molecule?
> >
> > Every molecule with Tversky similarity \< 1.0 on the x-axis is novel (not an exact match). We show the scatterplot to demonstrate that high-scoring molecules are not merely close variations of known molecules. There is no positive correlation between similarity and docking score. We agree that the correlation coefficient may distract from the main message but thought this was the best summary statistic; we welcome the reviewer's suggestions on how to present this data more effectively.

---

### Author Response · Authors · 2025-11-28

We thank the reviewers for their thoughtful feedback. We have added several additional experiments and edited the paper based on your comments:

- **Feedback alignment analysis** (GXdf, Ckan): LLM judge evaluates whether generated molecules follow true vs. scrambled feedback (81% win rate)
- **Binary-only ablation** (MZQt): Feedback Descent with preferences but no rationale text, directly testing whether textual feedback adds value beyond binary signals
- **Random feedback ablation** (MZQt, GXdf): Same algorithm with shuffled rationales, testing whether feedback content matters
- **Noisy feedback ablation** (jREt): 25%, 50%, 100% corrupted rationales, testing robustness to feedback quality
- **TextGrad comparison** (GXdf): Direct comparison on all 6 DOCKSTRING targets
- **Error bars** (GXdf): 3 random seeds across all tasks
- **Computational cost breakdown** (GXdf): Wall-clock time, API calls, and dollar costs
- **Theory section** (MZQt): We have now moved the formal propositions to the appendix and replaced Section 3 with a concise intuition paragraph in the methods section.

Major changes are marked in red. We address specific concerns in individual responses below.

---

### Note · Program_Chairs · 2026-01-17
**Submission Desk Rejected by Program Chairs**

The following references in this submission do not refer to real documents and/or have major errors in bibliographic information:

 Seonghyeon Kim, Sukmin Cho, Doyoung Kim, Sejin Kim, Chacha Chen, Ekaterina Kochmar, Hwajung Hong, and Alice Oh. Help me think: A simple prompting strategy for non-experts to create customized content with models. arXiv preprint arXiv:2208.08232, 2023.